# First Results from the Prospective German Registry for Childhood Glaucoma: Phenotype–Genotype Association

**DOI:** 10.3390/jcm11010016

**Published:** 2021-12-21

**Authors:** Julia V. Stingl, Stefan Diederich, Heidi Diel, Alexander K. Schuster, Felix M. Wagner, Panagiotis Chronopoulos, Fidan Aghayeva, Franz Grehn, Jennifer Winter, Susann Schweiger, Esther M. Hoffmann

**Affiliations:** 1Department of Ophthalmology, University Medical Center of the Johannes Gutenberg University Mainz, 55131 Mainz, Germany; julia.stingl@unimedizin-mainz.de (J.V.S.); hdiel@students.uni-mainz.de (H.D.); alexander.schuster@uni-mainz.de (A.K.S.); felix.wagner@unimedizin-mainz.de (F.M.W.); panagiotis.chronopoulos@unimedizin-mainz.de (P.C.); dr.aghayeva@gmail.com (F.A.); Grehn_F@ukw.de (F.G.); 2Institute for Human Genetics, University Medical Center of the Johannes Gutenberg University Mainz, 55131 Mainz, Germany; stefan.diederich@unimedizin-mainz.de (S.D.); jennifer.winter@unimedizin-mainz.de (J.W.); susann.schweiger@unimedizin-mainz.de (S.S.); 3National Centre of Ophthalmology Named after Academician Zarifa Aliyeva, Baku AZ1114, Azerbaijan

**Keywords:** childhood glaucoma, primary congenital glaucoma, CYP1B1, molecular genetic analysis, genotype phenotype correlation, consanguinity

## Abstract

Childhood glaucoma is a heterogeneous disease and can be associated with various genetic alterations. The aim of this study was to report first results of the phenotype–genotype relationship in a German childhood glaucoma cohort. Forty-nine eyes of 29 children diagnosed with childhood glaucoma were prospectively included in the registry. Besides medical history, non-genetic risk factor anamnesis and examination results, genetic examination report was obtained (23 cases). DNA from peripheral blood or buccal swab was used for molecular genetic analysis using a specific glaucoma gene panel. Primary endpoint was the distribution of causative genetic mutations and associated disorders. Median age was 1.8 (IQR 0.6; 3.8) years, 64% participants were female. Secondary childhood glaucoma (55%) was more common than primary childhood glaucoma (41%). In 14%, parental consanguinity was indicated. A mutation was found in all these cases, which makes consanguinity an important risk factor for genetic causes in childhood glaucoma. CYP1B1 (30%) and TEK (10%) mutations were found in primary childhood glaucoma patients. In secondary childhood glaucoma cases, alterations in CYP1B1 (25%), SOX11 (13%), FOXC1 (13%), GJA8 (13%) and LTBP2 (13%) were detected. Congenital cataract was associated with variants in FYCO1 and CRYBB3 (25% each), and one case of primary megalocornea with a CHRDL1 aberration. Novel variants of causative genetic mutations were found. Distribution of childhood glaucoma types and causative genes was comparable to previous investigated cohorts. This is the first prospective study using standardized forms to determine phenotypes and non-genetic factors in childhood glaucoma with the aim to evaluate their association with genotypes in childhood glaucoma.

## 1. Introduction

Glaucoma in children is rare. The incidence widely varies depending on the origin of the cohort and the classification of the type of glaucoma (see childhood glaucoma research network, CGRN [1]). Previous reports from western countries found incidences of primary congenital glaucoma (PCG) ranging from 1:10,000 to 1:40,000 births [2,3] whereas it is higher in countries where consanguine marriage is more common. Thus, in Saudi Arabia the incidence was reported 1:2500 [4], and the highest incidence of 1:1250 was found in a cohort of Slovakian Romani people [5].

This indicates the high impact of inheritance in childhood glaucoma. Hereditary cases account for 10 to 40% in PCG [6]. The most frequently mutated gene in PCG is CYP1B1, which is located on chromosome 2p22-p21 (locus GLC3A). CYP1B1 associated PCG is mostly inherited as an autosomal recessive trait with variable penetrance and a slight bias to males [7,8]. Furthermore, MYOC, LTBP2 and TEK/ANGPT1 mutations have been identified in PCG patients. Juvenile open-angle glaucoma is associated with MYOC alterations in up to 36% and with CYP1B1 in 2% [9,10]. Moreover, some of the secondary glaucoma types show a high heritability. In about 40% of patients with Axenfeld–Rieger anomaly (ARA), which is associated with glaucoma in 50 to 75%, either FOXC1 or PITX2 mutations are found [11,12,13]. Interestingly, FOXC1-associated glaucoma generally has an earlier onset than PITX2 related ARA [14], but glaucoma in PITX2 patients seems to be more severe and more difficult to treat [11]. Peter’s anomaly is genetically heterogeneous and frequently occurs de novo [8]. Aniridia, however, is nearly always caused by PAX6 variants, which has a high penetrance but variable expression and is found in one parent in about two thirds of cases [8,15].

To date, there is sparse knowledge about the genetic and nongenetic causes for childhood glaucoma in Germany, we are in the process of establishing a nationwide registry for childhood glaucoma. This paper reports first results of the relationship between phenotype and genotype in our clinical pilot study cohort.

## 2. Methods

### 2.1. Study Sample

For this prospective pilot study, 29 children attending the Childhood Glaucoma Center of the Department of Ophthalmology, University Medical Center of the Johannes Gutenberg-University Mainz, Germany, were recruited during the period from December 2018 to December 2019. The pilot study was conducted to examine the feasibility of a German registry for childhood glaucoma and to test the designed questionnaires and examination forms (see below for detailed description). The registry aims to include all cases of childhood glaucoma in Germany to evaluate epidemiological numbers, genetic and non-genetic risk factors, clinical characteristics and treatment success rates. The pilot study was successfully completed and a nation-wide expansion with a multicentric approach is currently in progress. The present study is a subanalysis concentrating on the description of a phenotype–genotype association.

Inclusion criteria were age <18 years and at least one eye diagnosed with glaucoma and classified on the basis of the criteria of the Childhood Glaucoma Research Network (CRGN) as shown in Table 1 [1,8]. Exclusion criterion was a language barrier.

Written informed consent was obtained either from the parents/caregivers or from both parents/caregivers and patients if they were old enough to understand the purpose of the study. The study adhered to the tenets of the Declaration of Helsinki. Approval of the local ethics committee (Ethics Commission of the State Chamber of Physicians of Rhineland-Palatinate, reference no. 837.496.16 (10184), date of approval 17 July 2017) and the local data safety commission were obtained.

### 2.2. Questionnai Res

Two questionnaires were completed by parents/caregivers and, if old enough, by the patients themselves. The first questionnaire enquired the medical history (age, date/place where the diagnosis of glaucoma was first suspected, date of first examination, affected eye, prior medication and surgeries, other diseases and medications), the second questionnaire comprised the gestational history (parental age, parental consanguinity, family history of childhood glaucoma, natural conception or in-vitro fertilization, birth weight and height, gestational age, risk factors during pregnancy such as alcohol, tobacco, drugs, medications and infections, and maternal folic acid intake).

### 2.3. Clinical Examination

Clinical examination was conducted in 24 cases either in general anesthesia (examination under anesthesia, EUA) or at the slit lamp and followed the standardized protocol normally used for examinations in the Mainz Childhood Glaucoma Center [16]. In EUA, intraocular pressure (IOP) was measured by both Perkins applanation tonometry (model Mk2, Haag-Streit Holding, Köniz, Switzerland) and iCare^®^ Pro rebound tonometry (iCare Finland Oy, Vantaa, Finland), at slit lamp examination by Goldmann applanation tonometry. Objective refraction (Retinomax K-plus2, Nikon Inc., Tokyo, Japan or Nidek AR1s, Nidek Co., Tokyo, Japan), corneal curvature (Retinomax K-plus2, Nikon Inc., Japan or Zeiss IOL Master 800, Carl Zeiss Meditec AG, Jena, Germany), corneal diameters (horizontal and vertical, by calipers), central corneal thickness (CCT; Tomey-AL-2000, Tomey, Nürnberg, Germany or 4Optics OCP Pachymeter, Heidelberg Enigneering, Heidelberg, Germany) and axial length (Tomey-AL-2000, Tomey, Nürnberg, Germany or Zeiss IOL Master 800) were measured. Anterior and posterior segments and chamber angle were examined by experienced childhood glaucoma experts (FG, PC and EMH).

### 2.4. Genetic Examination

Genetic examination was carried out in 23 cases. In 22 cases, peripheral blood was used for molecular genetic analysis in a whole exome sequencing approach using NGS technology. Therefore, the Agilent SureSelectXT Human All Exon V7 kit (Agilent, Santa Clara, CA, USA) was used to enrich all human exons, and 2 × 126 bp paired end sequencing was performed on an Illumina NextSeq500 (Illumina NextSeq 500/550 High Output Kit v2.5 (300 cycles)). Alignment was performed with BWA version 0.7.17-r1188 against the human reference genome HG19, and GATK version 4 was employed for variant calling. 97% of all targeted exons were covered by at least 10 reads. All variants were annotated with data from 1000 genomes (phase3), COSMIC (version 81-92), ClinVar (version 201706-202012), ESP (version 20141103), HGMD-PRO (version 20174), dbSNP (version 150-154), gnomAD (version 170228-r2.1), Polyphen (version 2.2.2), SIFT (version 5.2.2), OMIM (download from August 2018–April 2021), dbNSFP (version 2.9.3-4.0) and others using the Ensembl Variant Effect Predictor (VEP) version 94 to 104. A virtual panel of 24 genes related with childhood glaucoma was assembled in cooperation with the local Institute of Human Genetics (Mainz Childhood Glaucoma Gene Panel, Table 2) and analysed in the patients’ datasets. When no mutation was found but clinical findings suggested a syndromal disease, bioinformatic analysis was extended by a (virtual) ocular gene panel with up to 506 genes (full list available under https://www.unimedizin-mainz.de/humangenetik/diagnostik/gen-panels-und-einzelgendiagnostik/augenerkrankungen.html, accessed on 16 December 2021). One case was already diagnosed with FOXC1-assiociated Axenfeld Rieger-syndrome at the time of presentation; the FOXC1 mutation was confirmed in a buccal swab sample (panel contained FOXC1, CYP1B1 and PITX2). Datasets of four children with glaucoma following cataract surgery were analysed with a panel for non-syndromal cataract comprising 23 genes (Table 2). If a genetic anomaly was found in a patient, the parents were tested for the concerning alteration.

### 2.5. Statistics

Statistical analysis was conducted with IBM SPSS Statistics 27. For normally distributed variables, mean and standard deviation, for other variables median and interquartile range were calculated. Frequencies are reported in absolute and relative numbers (%). Figures were created with Microsoft Excel.

## 3. Results

A total of 49 eyes from 29 children with manifest or suspected diagnosis of primary or secondary congenital glaucoma (Table 3) were included in this pilot study. One patient (3%) with suspected childhood glaucoma was diagnosed with isolated megalocornea without glaucomatous damage. The median age was 1.8 (IQR 0.6; 3.8) years. 18 (64%) participants were female, 22 (76%) were born in Germany. Other countries of origin were east European countries (*n* = 4) or Asian countries (*n* = 3). The proportion of German origin was lower in the parents (51.7%). Diagnosis was bilateral in 58%. Consanguinity was indicated in 14%. A family history of childhood glaucoma was found in 21%. Baseline data are shown in Table 3.

Table 3 shows the distribution of childhood glaucoma types. Primary childhood glaucoma accounted for 41% (12 cases), namely 11 cases of PCG and one case of JOAG. Secondary childhood glaucoma comprised 55%. Nonacquired ocular anomalies were the most frequent etiology of secondary childhood glaucoma (31%), and Peters anomaly was the most common condition related with a secondary glaucoma (78% of non-acquired ocular anomalies, 24% of all cases). Glaucoma following cataract surgery occurred in 14% (4 cases).

Of 29 children included in this evaluation, 23 children (79%) underwent a molecular genetic analysis. For 6 children, examination was refused either by caregivers or insurance. Genetic findings in relation to clinical diagnosis are summarized in Figure 1.

In primary childhood glaucoma cases (*n* = 10), which comprised PCG and JOAG, the most common mutation was CYP1B1 (3 cases; 30%). One PCG case was associated with a mutation in TEK (10%). In 6 cases (60%), no mutation was found, for two cases no results were obtained as examination was refused by insurance.

The secondary childhood glaucoma cases showed diverse phenotypes and genotypes (*n* = 8): Peters anomaly was linked with CYP1B1 mutations in two cases (25%), no mutation was found in one case (13%). A SOX11 variant was identified in one patient (13%) with Coffin–Siris syndrome [17]. Axenfeld–Rieger anomaly was associated with a FOXC1 mutation in one case (13%), Sclerocornea was seen in a patient with a GJA8 alteration (13%) and Weill–Marchesani syndrome was diagnosed in a patient presenting with an LTBP2 change (12.5%). One patient showed childhood glaucoma secondary to Sturge–Weber syndrome (13%), but no mutation was found. In four cases, no genetic analysis was conducted because parents did not consent to it.

Glaucoma following cataract surgery (*n* = 4) was linked to a FYCO1 mutation in one case (25%) and to CRYBB3 in another (25%); two more patients (50%) did not show any genetic aberrations.

A mutation in the X-chromosomal gene CHRDL1 was identified in a patient with primary megalocornea but not glaucoma.

Table 4 summarizes some key clinical parameters comparing the number of cases with mutation and without mutation identified. In all patients with indicated parental consanguinity a mutation was found. A family history of childhood glaucoma seemed more frequent in the mutation groups, especially in secondary childhood glaucoma and post-cataract surgery cases. Interestingly, 11/12 mutations could be detected in at least one parent, which means that in the presented sample only one *de-novo* mutation was found (SOX11-related Coffin–Siris syndrome [17]). Bilateral disease seemed to be more common in the mutation groups. Both birth weight and gestational age appeared lower when no mutation was found.

Table 5 summarizes the genetic variants with sequence and resulting protein alterations in the cohort. Two novel variants in CYP1B1-associated PCG (c.80T>C and c.1345_1347delGATinsAC), two novel variants in LTBP2-associated Weill–Marchesani syndrome (c.4067G>A and c.5276G>C), one novel variant in FYCO1-associated congenital cataract (c.265C>T) and one novel variant in X-linked megalocornea (c.1002_1003del) were found. Two of these novel variants are not described in literature but listed in the dbSNP database (dbSNP identifier are listed instead of literature reference).

## 4. Discussion

Childhood glaucoma is a heterogeneous disease with various etiologies. The rarity of this pathology makes it difficult to find evidence-based therapies for the several different childhood glaucoma types. Causative non-genetic risk factors are sparsely studied, and genetic alterations do not explain all cases of childhood glaucoma [7].

Since genetic screening methods became more efficient and affordable during the last two decades, several studies regarding the relation of genotype and phenotype have been conducted. The Australian and New Zealand Registry of Advanced Glaucoma (ANZRAG) analysed the genotypes of a large childhood glaucoma registry but did not include cases of glaucoma following cataract surgery [26]. Several cohorts investigated the impact of mutations in single genes on clinical parameters or success of surgery, but merely included distinct mutations such as CYP1B1, MYOC or LTBP2 in PCG [27,28,29,30,31,32,33,34,35,36,37,38,39] or focused on individual syndromes possibly leading to secondary childhood glaucoma such as Axenfeld–Rieger anomaly [11,40] or Aniridia [41]. However, to ensure an optimal supply, it is important to compare influences of genetic mutations on clinical parameters of all childhood glaucoma categories.

This study is a subanalysis of the pilot study cohort in a German childhood glaucoma registry with the aim to examine the phenotype–genotype correlations in the different types of childhood glaucoma and their related genetic alterations in the German Registry for Childhood Glaucoma.

In this study, secondary childhood glaucoma was seen slightly more often than primary childhood glaucoma types. In earlier studies we had described secondary glaucoma less than 50% of all glaucoma cases in our population [16]. This discrepancy is probably due to the fact that glaucoma centers in Germany are rare and complex cases, which are generally secondary glaucoma cases, are referred. However, if the CGRN classification was applied, PCG was the most frequent childhood glaucoma (38%), prior to glaucoma associated with non-acquired ocular anomalies (31%), glaucoma following cataract surgery (14%), glaucoma associated with non-acquired systemic disease or syndrome (10%) and JOAG (3%). These findings are similar with previous studies, though the distribution of secondary glaucoma types slightly varies, possibly due to different methods or selection biases. In these studies, PCG occurred with a frequency of 32 to 58%, JOAG in 0.4 to 19%, glaucoma associated with non-acquired ocular anomalies in 5 to 18%, with non-acquired systemic disease in 1 to 12%, following cataract surgery in 1 to 13% and with acquired conditions in 1 to 30% of cases in prior childhood glaucoma cohorts [2,26,42,43,44,45]. However, cases of glaucoma with acquired conditions were not represented in our sample.

Male gender was less frequent than female gender in a 1:1.6 ratio, in contrast to previous studies showing a slightly [26,42,44,45] or even clearly [43,46] higher prevalence in males. This might be explained by the small number of participants in our pilot study cohort, which therefore might not be representative. Higher male:female ratios were also found for PCG in previous studies. Interestingly, an equal or higher prevalence of female gender was found in CYP1B1 related PCG [26,47,48].

Consanguinity was affirmed in 14% of the families in this study. In all cases associated with parental consanguinity, we found a causative genetic mutation. This fits well with the higher prevalence of CYP1B1 in PCG patients in countries where consanguinity is more common [27]. In this sample, in 11 of 12 cases featuring causative genetic variants, these variants were also detected in at least one of their parents, implying that only one of the patients had a de-novo mutation. Thus, it can be assumed that genetic alterations leading to childhood glaucoma are inherited in a high proportion. Moreover, it emphasizes the impact of consanguinity as risk factor for childhood glaucoma.

The genetic examination yielded a mutation in one third of PCG patients. The most common mutation was CYP1B1 (cytochrome P450 family 1 subfamily B member 1), which accounted for 30% of PCG cases. It was also the predominant mutation in PCG in previous registry studies [26,49]. A higher frequency of CYP1B1 mutations was seen in cohorts from Turkey [49], Saudi Arabia (75%) [27], Morocco (48%) [28], India (44%), and Brazil (44%) [31], whereas the frequency was lower in Chinese (17%) [30] and Australian/New Zealand patients (16%) [26]. The different frequency rates may be caused by different rates of consanguinity—the Saudi cohort reported consanguinity of 69%, which is almost five times as high as seen in our study. CYP1B1 encodes a B subfamily protein of cytochrome P450 1, which is expressed in several tissues of the human body and seems to be involved in development of trabecular meshwork [7]. Although CYP1B1-knockout mice showed similar alterations of trabecular meshwork and Schlemm’s canal as found in PCG patients, the exact molecular mechanism is not entirely understood [50].

We identified one PCG patient with an inherited mutation in TEK (TEK receptor tyrosine kinase, 10%). The ANZRAG registry reported TEK mutations in 6% of PCG patients [26]. Another cohort of PCG patients that did not feature any mutations in CYP1B1, MYOC, LTBP2 or FOXC1, showed a TEK frequency of 5% and suggested an autosomal dominant inheritance with variable expressivity and incomplete penetrance [51]. TEK is a receptor tyrosine kinase highly expressed in Schlemm’s canal endothelial cells. By binding its ligand ANGPT1, a vascular growth factor, formation of vacuoles in the endothelial cells of trabecular meshwork is mediated to improve outflow of aqueous humor. Further, activation of TEK/ANGPT1 cascade plays a pivotal role in maintenance of trabecular meshwork structures [7].

Further common mutations associated with PCG are MYOC, LTBP2, ANGPT1, COL1A1 and FOXC1 [7]. ANZRAC found mutations in CPAMD8 in 4%, FOXC1 in 4%, and ANGPT1 in 1% of PCG cases [26], a Turkish cohort FOXC1 mutations in 4% [49].

The one JOAG case did not show genetic anomalies.

Genotyping of secondary childhood glaucoma showed a more diverse picture and the highest rate of positive genetic examination results among all glaucoma types. This was similar in the ANZRAG registry, where the highest diagnostic yield was found in glaucoma associated with non-acquired ocular anomalies [26].

In the category of non-acquired ocular anomalies CYP1B1 was found in 25% and accounted for 2 of 7 Peter’s anomaly patients. Peter’s anomaly is a genetical heterogeneous syndrome. Aside from CYP1B1, which seems to be the most frequent mutation in Peter’s anomaly [52,53], other causative genes are FOXC1, PITX2, PAX6, FOXE3, PITX3 and B3GLCT [54]. The ANZRAG reported one case of Peter’s anomaly with a PITX2 mutation [26]. The pathomechanism seems to be an incomplete separation of the lens vesicle from the surface ectoderm tissue, which causes iridocorneal and lenticulocorneal adhesions [55].

One patient showed a mutation in FOXC1 and presented with Axenfeld–Rieger anomaly (ARA). FOXC1 and PITX2 are the most relevant genes in ARA and account for up to 40 to 63% of ARA cases [23,54]. ARA leads to glaucoma in 50 to 75%, which is believed to result from abnormal development of the angle [12]. FOXC1 (forkhead box protein C1) and PITX2 (pituitary homebox 2) are transcription factors regulating embryogenesis, cell proliferation and differentiation in neural crest cells forming ciliary body, iris and trabecular meshwork [13,56]. Their malfunction contributes to anterior segment dysgenesis [54]. Since these transcription factors are not only expressed in ocular tissues, but also in fetal and adult heart, brain and kidneys, extraocular comorbidities such as congenital heart defects or hearing loss are common [13].

We found a GJA8 (gap junction protein alpha 8) mutation in one patient with the clinical diagnosis of sclerocornea. Sclerocornea is familial in about 50% of cases and inheritance is possible in both autosomal dominant and recessive manners [8]. GJA8 encodes gap junction proteins in the lens and influences the lens fiber hemostasis. Therefore, the main clinical finding reported in the literature is cataract, but also anterior segment dysgenesis such as sclerocornea and microphthalmia are frequently described [54], which were also seen in our patient.

The group of non-acquired systemic diseases included a Weill–Marchesani case with LTBP2 mutation, a patient with Coffin–Siris syndrome with SOX11 mutation [17], and one case with Sturge–Weber syndrome.

Weill–Marchesani syndrome is a connective tissue disorder leading to microspherophakia and consecutively to ectopia lentis due to lax or absent zonules. Glaucoma mainly derives from angle closing, and patients might also show anterior peripheral synechiae from iridocorneal approximation. Common related genetic mutations are ADAMTS10, ADAMTS17, FBN1 and LTBP2. LTBP2 can also cause primary megalocornea and lead to an incorrect diagnosis [8,57]. LTBP2 (late transforming growth factor-beta binding protein) is highly expressed in trabecular meshwork and ciliary processes regulating formation of ciliary microfibrils. The microfibrils consist of fibrillins 1 and 2, which are encoded by FBN1 and FBN2. ADAMTS10 and ADAMTS17 are extracellular matrix proteases influencing the extracellular matrix and stirring fibrillin cleavage [54,58]. Previous cohorts rarely reported Weill–Marchesani syndrome related cases; the ANZRAG described one case that was diagnosed correctly only after genetical examination (ADAMTS17) [26].

We recently reported a case of SOX11 (sex-determining region Y-related high-mobility-group box transcription factor 11)-related Coffin–Siris syndrome with the first observation of childhood glaucoma [17] presenting with maximal intraocular pressure of 43 mmHg, microcornea with central opacity, aniridia and cataract. Few cases of anterior segment dysgenesis are described in the literature, which is suggested to occur as a result of incomplete separation of the lens vesicle from the surface ectoderm, similar to Peter’s anomaly [59].

Contrary to the latter two depicted diseases, Sturge–Weber syndrome is more commonly reported in childhood glaucoma cohorts [2,26,42,43,44,46]. Sturge–Weber syndrome belongs to the group of phacomatoses and causes secondary glaucoma in 30% with an increased risk in eyes with cutaneous capillary malformation in the upper eyelid and choroidal hemangioma [8]. Glaucoma may result from primary goniodysgenesis, but also from reduced aqueous outflow caused by an elevation of episcleral venous pressure [8]. Somatic mosaic mutations in GNAQ and PI3K have been found in patients with Sturge–Weber syndrome. Germline mutations, however, may only be prognostic factors [60,61]. As these mutations can only be identified by biopsy, neither this nor other childhood glaucoma registries for genetic examination found genetic alterations in Sturge–Weber patients [26].

The sample further comprised four cases of glaucoma following cataract surgeries. Genetic examination revealed a CRYBB3 (crystallin beta b3) mutation in one, and a FYCO1 (FYVE and coiled-coil domain containing 1) mutation in another patient. Congenital cataract is inherited in about 22% of cases [62]. CRYBB3 encodes a crystallin, which is the main protein type in the human lens and is responsible for its transparency and refractive index due to their firm packing. Alterations in the crystallin genes and protein structures cause opacities [62]. FYCO1 is a membrane protein involved in autophagy, which is an important process during lens development and lens fiber differentiation to create a transparent organelle free zone [63]. Mutations in crystallin genes are responsible for around half, membrane proteins for a quarter of inherited cataract cases [62]. One case of glaucoma following cataract surgery without mutation in the molecular panel analysis was associated with trisomy 21, which is a possible causative chromosomal aberration for congenital cataract [64].

One patient with suspected congenital glaucoma revealed CHRDL1 associated X-linked megalocornea. CHRDL1 (chordin-like 1) encodes for the protein ventropin, which—if deficient—leads to excessive corneal proliferation resulting in thinned and enlarged corneal dimensions without elevation of IOP [65].

The presented study had some limitations. Due to the small cohort size, it was not possible to determine the prevalence of mutations in specific glaucoma types and an analysis of the relation between genotype and severity of disease would not have enough statistical strength.

Not all the known genetic variants for PCG were examined in the regular gene panel. For example, ANGPT and COL1A1 were not tested. Thus, the proportions of mutational childhood glaucoma could differ from our findings.

Cases of glaucoma associated with acquired conditions were not represented in our pilot study cohort because of a low incidence during the recruitment time frame. The aim of the registry is to include all cases and all types of childhood glaucoma presenting to the participating childhood glaucoma centers.

## 5. Conclusions

We present a subanalysis of the pilot study cohort to the prospective nationwide German Registry for Childhood Glaucoma to analyse the phenotype–genotype correlation in childhood glaucoma. A diverse set of mutations linked to heterogenous clinical findings was seen. Novel variants of causative genetic mutations were found. Further patients will be included, and nationwide expansion is currently in progress. The analysis showed a similar distribution of childhood glaucoma types and their underlying genes in Germany as in previous childhood glaucoma cohorts. It further underlines the important influence of the risk factor consanguinity.

## Figures and Tables

**Figure 1 jcm-11-00016-f001:**
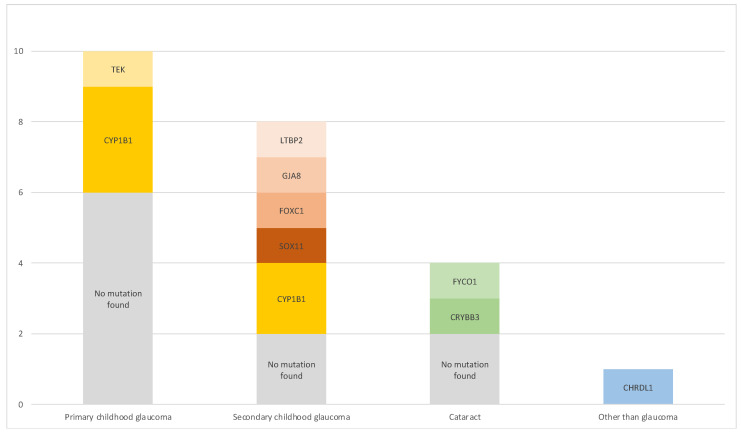
Phenotype–genotype relationship. Cases with missing results were excluded.

**Table 1 jcm-11-00016-t001:** Childhood Glaucoma Research Network (CGRN) diagnosis criteria and classification [1].

**Diagnosis Criteria**
IOP > 21 mmHg Optic nerve cupping Increasing cup-to-disc ratio Asymmetry ≥ 0.2 Focal thinning Corneal findings Haab striae Diameter ≥ 11 mm in newborns, >12 mm in children <1 year, >13 mm any age Corneal edema Axial length increase Myopic shift linked with exceeding ocular globe growth Visual field defects Reproducible and compatible with glaucomatous damage and no other existing reason for defect Childhood glaucoma: ≥2 criteria Glaucoma suspect: IOP > 21 mmHg on 2 different occasions or ≥1 other criteria
**Classification**
Primary congenital glaucoma Neonatal onset (≤1 month) Infantile onset (<1–24 months) Late onset (<2 years) Juvenile open angle glaucoma Glaucoma associated with non-acquired ocular anomalies Glaucoma associated with non-acquired systemic disease or syndrome Glaucoma with acquired conditions Glaucoma following cataract surgery Congenital idiopathic cataract Congenital cataract associated with ocular anomalies/systemic disease Acquired cataract

**Table 2 jcm-11-00016-t002:** Mainz Childhood Glaucoma Gene Panel.

Glaucoma	Cataract
CNTNAP2, COL4A1, COL11A1, CYP1B1, FOXC1, FOXE3, LMX1B, LOXL1, LTBP2, MAF, MYOC, OPA1, OPA3, OPTN, PAX6, PITX3, PLEKHA7, ST18, TBK1, TEK, TMEM126A, WDR36	AGK, BFSP1, BFSP2, CRYAA, CRYAB, CRYBA1, CRYBA4, CRYBB1, CRYBB2, CRYBB3, CRYGC, CRYGD, CRYGS, EPHA2, FOXE3, FYCO1, GCNT2, GJA3, GJA8, HSF4, LIM2, MIP, PITX3

**Table 3 jcm-11-00016-t003:** Demographic characteristics.

Baseline Data	
Age	
Median	1.8 (IQR 0.6; 3.8) years
Sex	
Female	18 (62%)
Male	11 (38%)
Laterality	
Unilateral	12 (41%)
Bilateral	17 (59%)
Childhood glaucoma	
Primary childhood glaucoma	12 (41%)
Primary congenital glaucoma	11 (3%)
Juvenile open-angle glaucoma	1 (3%)
Secondary childhood glaucoma	16 (55%)
Glaucoma associated with non-acquired ocular anomalies	9 (31%)
Peters anomaly	7 (24%)
Axenfeld–Rieger anomaly	1 (3%)
Sclerocornea	1 (3%)
Glaucoma associated with non-acquired systemic syndromes or diseases	3 (10%)
Sturge–Weber syndrome	1 (3%)
Coffin–Siris syndrome	1 (3%)
Weill-Marchesani syndrome	1 (3%)
Glaucoma associated with acquired conditions	0 (0%)
Glaucoma following cataract surgery	4 (14%)
Glaucoma suspect	
Megalocornea	1 (3%)
Country of birth	
Germany	22 (76%)
East European countries	4 (14%)
Asia	3 (10%)
Consanguinity	
Yes	4 (14%)
No	25 (86%)
Unknown	0 (0%)
Family history of childhood glaucoma	
Yes	6 (21%)
No	22 (76%)
Unknown	1 (3%)

**Table 4 jcm-11-00016-t004:** Comparison of non-genetic risk factors in patients with and without found mutations.

	Cases, n (%)	Male Gender, n (%)	Consanguinity, n (%)	Family History, n (%)	Bilateral, n (%)	Birth Weight (g)	Gestational Age (Weeks)
Primary childhood glaucoma							
Mutation	4/12 (33.3%)	1/4 (25%)	1/4 (25%)	0/4 (0%)	4/4 (100%)	3725 [2654; 4038]	40.9 [37.9; 41.0]
No mutation found	6/12 (50%)	3/6 (50%)	0/6 (0%)	1/6 (17%)	5/6 (83%)	3340 [2623; 3878]	39.4 [38.6; 40.3]
Secondary childhood glaucoma (except glaucoma following cataract surgery)							
Mutation	6/12 (50%)	3/6 (50%)	2/6 (33%)	3/5 (60%)	6/6 (100.%)	3530 [3170; 3850]	39.8 [38.4; 41.5]
No mutation found	2/12 (17%)	0/2 (0%)	0/2 (0%)	0/2 (0%)	1/2 (50%)	1873 [565; 1873]	32.3 [24.1; 32.3]
Glaucoma following cataract surgery							
Mutation	2/4 (50%)	1/2 (50%)	1/2 (50%)	2/2 (100%)	1/2 (50%)	2540 [1970; 2540]	38.6 [37.6; 38.6]
No mutation found	2/4 (50%)	0/2 (0%)	0/2 (0%)	0/2 (0%)	0/2 (0%)	2380 [2050; 2380]	38.5 [38.0; 38.5]

**Table 5 jcm-11-00016-t005:** Genetic variants in childhood glaucoma patients.

Patient	Phenotype	Gene	Location	Region	Function	Status	Sequence Alteration	Protein Alteration	Previous Literature
1	PCG	CYP1B1	2p22.2	Exon 2/Exon 2	Endogenous steroid metabolism	heterozygous	c.535delG/c.80T>C	p.(Ala179Argfs*18)/p.(Leu27Pro)	[18]/novel
6	PCG	CYP1B1	2p22.2	Exon 2	Endogenous steroid metabolism	homozygous	c.868dup	p.(Thr404SerfsTer30)	[19]
19	PCG	CY1B1	2p22.2	Exon 3/Exon 3	Endogenous steroid metabolism	heterozygous	c.1345_1347delGATinsAC/c.1200_1209dup	p.(Asp449ThrfsTer8)	novel/[20]
10	PCG	TEK	9p21.2	Exon 14	Formation and hemostasis of TM/SC	heterozygous	c.2228G>C	p.(Gly743Ala)	[21]
28 ^†^	Peters anomaly	CYP1B1	2p22.2	Exon 3	Endogenous steroid metabolism	homozygous	c.1169G>A	p.(Arg390His)	[22]
29 ^†^	Peters anomaly	CYP1B1	2p22.2	Exon 3	Endogenous steroid metabolism	homozygous	c.1169G>A	p.(Arg390His)	[22]
4	Coffin–Siris syndrome	SOX11	2p25.2	Exon 1	Development of anterior segment	heterozygous	c.251G>T	p.(Gly84Val)	[17]
21	Axenfeld–Rieger syndrome	FOXC1	6p25.3	NA	Development of anterior Segment	heterozygous	NA (Deletion)	NA	[23]
20	Sclerocornea	GJA8	1q21.2	Exon 2	Corneal/lens gap junction protein	heterozygous	c.280G>C	p.(Gly94Arg)	[24]
22	Weill–Marchesani syndrome	LTBP2	14q24.3	Exon 28/35	ECM hemostasis in TM and ciliary processes	heterozygous	c.4067G>A/c.5276G>C	p.(Cys1356Tyr)/p.(Cys1759Ser)	rs778353798 ^‡^/novel
15	Congenital cataract	CRYBB3	22q11.23	Exon 5	Lens protein	heterozyous	c.466G>A	p.(Gly156Arg)	[25]
17	Congential cataract	FYCO1	3p21.31	Exon 4	Autophagy during lens development	homozygous	c.265C>T	p.(Arg89Cys)	rs141476300 ^‡^
14	X-linked Megalocornea	CHRDL1	Xq23	Exon 10	Control of corneal proliferation	hemizygous	c.1002_1003del	p.(Gln335LysfsTer3)	novel

Abbreviations: PCG, primary congenital glaucoma; CYP1B1, cytochrome P450 family 1 subfamily B member 1; TEK, TEK receptor tyrosine kinase; TM, trabecular meshwork; SC, Schlemm’s canal; SOX11, sex-determining region Y-related high-mobility-group box transcription factor 11; FOXC1, forkhead box C1; GJA8, gap junction protein alpha 8; LTBP2, latent transforming growth factor-beta-binding protein 2; ECM, extracellular matrix; CRYBB3, crystallin beta b3; FYCO1, FYVE and coiled-coil domain containing 1; CHRDL1, chordin-like 1. ^†^ Siblings. ^‡^ For variants listed in dbSNP but not in literature, dbSNP identifier were listed.

## Data Availability

The data presented in this study are available on request from the corresponding author. The data are not publicly available due to their containing information that could compromise the privacy of research participants.

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
