# Peer review of "First Results from the Prospective German Registry for Childhood Glaucoma: Phenotype–Genotype Association"

_jcm, 2021, doi:10.3390/jcm11010016_

Round 1
Reviewer 1 Report
Dr Hoffmann et coworkers from Germany have written a paper entitled “Ancillary Study to the Feasibility Study of the prospective German Registry for Childhood Glaucoma: Phenotype-Genotype Association”.
This article seems to be an early report of their prospective study concerning genetics of the childhood glaucoma in Germany. The aim of their study is interesting. The data base of their study is novel. Overall the (future) results of their research will give a remarkable impulse in understanding a rare and a challenging disease, childhood glaucoma.
The percent numbers should be given without decimals, otherwise this early report is accurate and well written. Waiting for further results.
Author Response
Answers to Reviewer 1
Dr Hoffmann et coworkers from Germany have written a paper entitled “Ancillary Study to the Feasibility Study of the prospective German Registry for Childhood Glaucoma: Phenotype-Genotype Association”.
This article seems to be an early report of their prospective study concerning genetics of the childhood glaucoma in Germany. The aim of their study is interesting. The data base of their study is novel. Overall the (future) results of their research will give a remarkable impulse in understanding a rare and a challenging disease, childhood glaucoma.
The percent numbers should be given without decimals, otherwise this early report is accurate and well written. Waiting for further results.
Our answer: We thank the reviewer for his/her positive review. We are looking forward to present further results. All percent numbers were changed as proposed.

Reviewer 2 Report
Some minor revisions:
Line 86: The formatting of the table content (font size, bullet points, alignment, etc.) makes the table confusing at least to non-clinical scientists. For table formatting please look at Aponte, Diehl, & Mohney (2010). Archives of ophthalmology, 128(4), 478-482.
Also, is “IOP < 21 mmHg” correct? Or should it be “IOP >21 mmHg”
Line 121: Mainz Childhood Glaucoma Gene Panel, Table 1 there is a typo; change to Table 2
Line 222: what does “subanalysis” mean here? Why not and what’s the full analysis?
Line 233: remove “following”
Line 307: revise the use of “latter” in the sentence
Line 324: revise the sentence
Line 161: Figure 1 seems redundant -- it just presents table 2 data.
A few other suggestions:
1- you can add more descriptions for other genes in the same way you did for CRYBB3, FYCO1, and CHRDL1 (in lines 331-343).
2- you can describe mutations/variants you found for each gene and maybe the potential molecular/physiological consequences. Maybe add a descriptive table that includes these genes, their function/localization, the mutation you identified, previous mutations identified, and citations of previous literature that studied these genes.
3- you can add some description of the whole-exome sequencing approach (line 119).
4- you may use a more concise title like "Prospective German Registry for Childhood Glaucoma: Phenotype-Genotype Association"
5- you may add more info about the status of the registry. Is it going to be a platform like ANZRAG?
Author Response
Answers to Reviewer 2
Some minor revisions:
Line 86: The formatting of the table content (font size, bullet points, alignment, etc.) makes the table confusing at least to non-clinical scientists. For table formatting please look at Aponte, Diehl, & Mohney (2010). Archives of ophthalmology, 128(4), 478-482.
Our answer: We thank the reviewer for this valuable hint. The table was separated in two sections, one for diagnosis criteria and another for classification (both following the childhood glaucoma research network [CGRN] guidelines). Classification of glaucoma based on gonioscopic findings (open-angle and angle-closure glaucoma) was removed for better clarity.
Also, is “IOP < 21 mmHg” correct? Or should it be “IOP >21 mmHg”
Our answer: We thank the reviewer for the careful perusal. This mistake was corrected.
Line 121: Mainz Childhood Glaucoma Gene Panel, Table 1 à there is a typo; change to Table 2
Our answer: Again, we are thankful for detecting this mistake. It was changed to “Table 2".
Line 222: what does “subanalysis” mean here? Why not and what’s the full analysis?
Our answer: We thank the reviewer for scrutinizing the term “subanalysis”. The description of the phenotype-genotype association was only one part within the scope of the registry. For further description of raised data see Methods, lines 93-117. The study protocol description and first results of the questionnaires and clinical variables is currently under review in another journal. We added an explaining sentence to the methods section, line 78-79:
“The present study is a subanalysis concentrating on the description of a phenotype-genotype association.”
Line 233: remove “following”
Our answer: We thank the reviewer for this correction. As “glaucoma following cataract surgery” is an established term, we instead removed “followed by” and replaced “prior to” in the previous sentence (lines 262-266):
“However, if the CGRN classification was applied, PCG was the most frequent childhood glaucoma (38 %), prior to glaucoma associated with non-acquired ocular anomalies (31 %), glaucoma following cataract surgery (14 %), glaucoma associated with non-acquired systemic disease or syndrome (10 %) and JOAG (3 %).”
Line 307: revise the use of “latter” in the sentence
Our answer: We are thankful for this style improvement. The sentence was corrected as follow (lines 353-355):
“Common related genetic mutations are ADAMTS10, ADAMTS17, FBN1 and LTBP2. LTBP2 can also cause primary megalocornea and lead to an incorrect diagnosis [1, 2].”
Line 324: revise the sentence
Our answer: We thank the reviewer for his/her valuable hint. The sentence was modified as follows (375-379):
“Somatic mosaic mutations in GNAQ and PI3K have been found in patients with Sturge-Weber syndrome. Germline mutations, however, may only be prognostic factors [3, 4].”
Line 161: Figure 1 seems redundant -- it just presents table 2 data.
Our answer: We thank the reviewer for this justified criticism. We removed figure 1.
A few other suggestions:
1- you can add more descriptions for other genes in the same way you did for CRYBB3, FYCO1, and CHRDL1 (in lines 331-343).
Our answer: We thank the reviewer for this great suggestion. We supplemented several explanations of genes/pathomechanisms as follows:
Lines 298-302: “CYP1B1 encodes a B subfamily protein of cytochrome P450 1, which is expressed in several tissues of the human body and seems to be involved in development of trabecular meshwork [5]. Although CYP1B1-knockout mice showed similar alterations of trabecular meshwork and Schlemm’s canal as found in PCG patients, the exact molecular mechanism is not entirely understood [6].”
Lines 307-312: “TEK is a receptor tyrosine kinase highly expressed in Schlemm’s canal endothelial cells. By binding its ligand ANGPT1, a vascular growth factor, formation of vacuoles in the endothelial cells of trabecular meshwork is mediated to improve outflow of aqeous humor. Further, activation of TEK/ANGPT1 cascade plays a pivotal role in maintenance of trabecular meshwork structures [5].”
Lines 333-339: “FOXC1 (forkhead box protein C1) and PITX2 (pituitary homebox 2) are transcription factors regulating embryogenesis, cell proliferation and differentiation in neural crest cells forming ciliary body, iris and trabecular meshwork [7, 8]. Their malfunction contribute to anterior segment dysgenesis [9]. Since these transcription factors are not only expressed in ocular tissues, but also in fetal and adult tissues of heart, brain and kidneys, extraocular comorbidities such as congenital heart defects or hearing loss are common [8].”
Lines 355-359: “LTBP2 (late transforming growth factor-beta binding protein) is highly expressed in trabecular meshwork and ciliary processes regulating formation of ciliary microfibrils. The microfibrils consist of fibrillins 1 and 2 wich are encoded by FBN1 and FBN2. ADAMTS10 and ADAMTS17 are extracellular matrix proteases interacting in remodeling of extracelluar matrix and fibrillin cleaving [9, 10].”
2- you can describe mutations/variants you found for each gene and maybe the potential molecular/physiological consequences. Maybe add a descriptive table that includes these genes, their function/localization, the mutation you identified, previous mutations identified, and citations of previous literature that studied these genes.
Our answer: This is an excellent idea to describe the found the genetic variants. We added table 5 to the manuscript:
Line 32 (Abstract): Novel variants of causative genetic mutations were found.
Lines 221-227: Table 5 summarizes the genetic variants with sequence and resulting protein alterations in the cohort. Two novel variants in CYP1B1-associated PCG (c.80T>C and c.1345_1347delGATinsAC), two novel variants in LTBP2-associated Weill-Marchesani syndrome (c.4067G>A and c.5276G>C), one novel variant in FYCO1-associated congenital cataract (c.265C>T) and one novel variant in X-linked megalocornea (c.1002_1003del) were found. Two of these novel variants are not described in literature but listed in the dbSNP database (dbSNP identifier are listed instead of literature reference).
Table 5. Genetic variants in childhood glaucoma patients
|
Patient |
Phenotype
|
Gene |
Location |
Region |
Function |
Status |
Sequence alteration |
Protein alteration |
Previous literature |
|
1 |
PCG |
CYP1B1 |
2p22.2 |
Exon 2/Exon 2 |
Endogenous steroid metabolism |
heterozygous |
c.535delG/c.80T>C |
p.(Ala179Argfs*18)/p.(Leu27Pro) |
[11]/novel |
|
6 |
PCG |
CYP1B1 |
2p22.2 |
Exon 2 |
Endogenous steroid metabolism |
homozygous |
c.868dup |
p.(Thr404SerfsTer30) |
[12] |
|
19 |
PCG |
CY1B1 |
2p22.2 |
Exon 3/Exon 3 |
Endogenous steroid metabolism |
heterozygous |
c.1345_1347delGATinsAC/c.1200_1209dup |
p.(Asp449ThrfsTer8) |
novel/[13] |
|
10 |
PCG |
TEK |
9p21.2 |
Exon 14 |
Formation and hemostasis of TM/SC |
heterozygous |
c.2228G>C |
p.(Gly743Ala) |
[14] |
|
28* |
Peters anomaly |
CYP1B1 |
2p22.2 |
Exon 3 |
Endogenous steroid metabolism |
homozygous |
c.1169G>A |
p.(Arg390His) |
[15] |
|
29* |
Peters anomaly |
CYP1B1 |
2p22.2 |
Exon 3 |
Endogenous steroid metabolism |
homozygous |
c.1169G>A |
p.(Arg390His) |
[15] |
|
4 |
Coffin-Siris syndrome |
SOX11 |
2p25.2 |
Exon 1 |
Development of anterior segment |
heterozygous |
c.251G>T |
p.(Gly84Val) |
[16] |
|
21 |
Axenfeld-Rieger syndrome |
FOXC1 |
6p25.3 |
NA |
Development of anterior Segment |
heterozygous |
NA (Deletion) |
NA |
[17] |
|
20 |
Sclerocornea |
GJA8 |
1q21.2 |
Exon 2 |
Corneal/lens gap junction protein |
heterozygous |
c.280G>C |
p.(Gly94Arg) |
[18] |
|
22 |
Weill-Marchesani syndrome |
LTBP2 |
14q24.3 |
Exon 28/35 |
ECM hemostasis in TM and ciliary processes |
heterozygous |
c.4067G>A/c.5276G>C |
p.(Cys1356Tyr)/p.(Cys1759Ser) |
rs778353798‡/novel |
|
15 |
Congenital cataract |
CRYBB3 |
22q11.23 |
Exon 5 |
Lens protein |
heterozyous |
c.466G>A |
p.(Gly156Arg) |
[19] |
|
17 |
Congential cataract |
FYCO1 |
3p21.31 |
Exon 4 |
Autophagy during lens development |
homozygous |
c.265C>T |
p.(Arg89Cys) |
rs141476300‡ |
|
14 |
X-linked Megalocornea |
CHRDL1 |
Xq23 |
Exon 10 |
Control of corneal proliferation |
hemizygous |
c.1002_1003del |
p.(Gln335LysfsTer3) |
novel |
Abbreviations: PCG, primary congenital glaucoma; CYP1B1, cytochrome P450 family 1 subfamily B member 1; TEK, TEK receptor tyrosine kinase; TM, trabecular meshwork; SC, Schlemm’s canal; SOX11, Sex-determining region Y-related high-mobility-group box transcription factor 11; FOXC1, forkhead box C1; GJA8, gap junction protein alpha 8; LTBP2, latent transforming growth factor-beta-binding protein 2; ECM, extracellular matrix; CRYBB3, crystallin beta b3; FYCO1, FYVE and coiled-coil domain containing 1; CHRDL1, chordin-like 1. *Siblings. ‡For variants listed in dbSNP but not in literature dbSNP identifier were listed.
Line 417: Novel variants of causative gentic mutations were found.
3- you can add some description of the whole-exome sequencing approach (line 119).
Our answer: We thank the reviewer for this suggestion. The description of whole-exome sequencing was supplemented as follows:
Lines 122-135: “In 22 cases, peripheral blood was used for molecular genetic analysis in a whole exome sequencing approach using NGS technology. Therefore, the Agilent SureSelectXT Human All Exon V7 kit (Agilent, Santa Clara, CA, USA) was used to enrich all human exons, and 2x126bp paired end sequencing was performed on an Illumina NextSeq500 (Illumina NextSeq 500/550 High Output Kit v2.5 (300 cycles)). Alignment was performed with BWA version 0.7.17-r1188 against the human reference genome HG19, and GATK version 4 was employed for variant calling. 97% of all targeted exons were covered by at least 10 reads. All variants were annotated with data from 1000 genomes (phase3), COSMIC (version 81 - 92), ClinVar (version 201706 - 202012), ESP (version 20141103), HGMD-PRO (version 20174), dbSNP (version 150 - 154), gnomAD (version 170228 - r2.1), Polyphen (version 2.2.2), SIFT (version 5.2.2), OMIM (download from August 2018 – April 2021), dbNSFP (version 2.9.3 – 4.0) and others using the Ensembl Variant Effect Predictor (VEP) version 94 to 104.”
4- you may use a more concise title like "Prospective German Registry for Childhood Glaucoma: Phenotype-Genotype Association"
Our answer: We thank the reviewer for this excellent suggestion. The title was changed into “First results from the prospective German Registry for Childhood Glaucoma: Phenotype-Genotype Association”.
5- you may add more info about the status of the registry. Is it going to be a platform like ANZRAG?
Our answer: This is a useful suggestion to improve the presentation of the registry’s goals. The methods part was supplemented as follows (lines 75-80):
“The registry aims at including all cases of childhood glaucoma in Germany to evaluate epidemiological numbers, genetic and non-genetic risk factors, clinical characteristics and treatment success rates. The pilot study was successfully completed and a nation-wide expansion with a multicentric approach is currently in progress. The present study is a subanalysis concentrating on the description of a phenotype-genotype association.”
References:
- Weinreb RN GA, Papadopoulos M, Grigg J, Freedman S: Childhood Glaucoma. The 9th Consensus Report of the World Glaucoma Association. Amsterdam: Kugler Publications; 2013.
- Marzin P, Cormier-Daire V, Tsilou E: Weill-Marchesani Syndrome. In: GeneReviews(®). edn. Edited by Adam MP, Ardinger HH, Pagon RA, Wallace SE, Bean LJH, Mirzaa G, Amemiya A. Seattle (WA): University of Washington, Seattle
Copyright © 1993-2021, University of Washington, Seattle. GeneReviews is a registered trademark of the University of Washington, Seattle. All rights reserved.; 1993.
- Shirley MD, Tang H, Gallione CJ, Baugher JD, Frelin LP, Cohen B, North PE, Marchuk DA, Comi AM, Pevsner J: Sturge-Weber syndrome and port-wine stains caused by somatic mutation in GNAQ. N Engl J Med 2013, 368(21):1971-1979.
- Nguyen V, Hochman M, Mihm MC, Jr., Nelson JS, Tan W: The Pathogenesis of Port Wine Stain and Sturge Weber Syndrome: Complex Interactions between Genetic Alterations and Aberrant MAPK and PI3K Activation. Int J Mol Sci 2019, 20(9).
- Ling C, Zhang D, Zhang J, Sun H, Du Q, Li X: Updates on the molecular genetics of primary congenital glaucoma (Review). Exp Ther Med 2020, 20(2):968-977.
- Doshi M, Marcus C, Bejjani BA, Edward DP: Immunolocalization of CYP1B1 in normal, human, fetal and adult eyes. Exp Eye Res 2006, 82(1):24-32.
- Gauthier AC, Wiggs JL: Childhood glaucoma genes and phenotypes: Focus on FOXC1 mutations causing anterior segment dysgenesis and hearing loss. Exp Eye Res 2020, 190:107893.
- Seifi M, Walter MA: Axenfeld-Rieger syndrome. Clin Genet 2018, 93(6):1123-1130.
- Ma AS, Grigg JR, Jamieson RV: Phenotype-genotype correlations and emerging pathways in ocular anterior segment dysgenesis. Hum Genet 2019, 138(8-9):899-915.
- Karoulias SZ, Beyens A, Balic Z, Symoens S, Vandersteen A, Rideout AL, Dickinson J, Callewaert B, Hubmacher D: A novel ADAMTS17 variant that causes Weill-Marchesani syndrome 4 alters fibrillin-1 and collagen type I deposition in the extracellular matrix. Matrix Biol 2020, 88:1-18.
- Cardoso MS, Anjos R, Vieira L, Ferreira C, Xavier A, Brito C: CYP1B1 gene analysis and phenotypic correlation in Portuguese children with primary congenital glaucoma. Eur J Ophthalmol 2015, 25(6):474-477.
- Souzeau E, Hayes M, Zhou T, Siggs OM, Ridge B, Awadalla MS, Smith JE, Ruddle JB, Elder JE, Mackey DA et al: Occurrence of CYP1B1 Mutations in Juvenile Open-Angle Glaucoma With Advanced Visual Field Loss. JAMA Ophthalmol 2015, 133(7):826-833.
- Chitsazian F, Tusi BK, Elahi E, Saroei HA, Sanati MH, Yazdani S, Pakravan M, Nilforooshan N, Eslami Y, Mehrjerdi MA et al: CYP1B1 mutation profile of Iranian primary congenital glaucoma patients and associated haplotypes. J Mol Diagn 2007, 9(3):382-393.
- Kabra M, Zhang W, Rathi S, Mandal AK, Senthil S, Pyatla G, Ramappa M, Banerjee S, Shekhar K, Marmamula S et al: Angiopoietin receptor TEK interacts with CYP1B1 in primary congenital glaucoma. Hum Genet 2017, 136(8):941-949.
- Stoilov I, Akarsu AN, Alozie I, Child A, Barsoum-Homsy M, Turacli ME, Or M, Lewis RA, Ozdemir N, Brice G et al: Sequence analysis and homology modeling suggest that primary congenital glaucoma on 2p21 results from mutations disrupting either the hinge region or the conserved core structures of cytochrome P4501B1. Am J Hum Genet 1998, 62(3):573-584.
- Diel H, Ding C, Grehn F, Chronopoulos P, Bartsch O, Hoffmann EM: First observation of secondary childhood glaucoma in Coffin-Siris syndrome: a case report and literature review. BMC Ophthalmol 2021, 21(1):28.
- Reis LM, Tyler RC, Volkmann Kloss BA, Schilter KF, Levin AV, Lowry RB, Zwijnenburg PJ, Stroh E, Broeckel U, Murray JC et al: PITX2 and FOXC1 spectrum of mutations in ocular syndromes. Eur J Hum Genet 2012, 20(12):1224-1233.
- Ma AS, Grigg JR, Prokudin I, Flaherty M, Bennetts B, Jamieson RV: New mutations in GJA8 expand the phenotype to include total sclerocornea. Clin Genet 2018, 93(1):155-159.
- Li D, Wang S, Ye H, Tang Y, Qiu X, Fan Q, Rong X, Liu X, Chen Y, Yang J et al: Distribution of gene mutations in sporadic congenital cataract in a Han Chinese population. Mol Vis 2016, 22:589-598.
